# The Role of Culinary Tourism in Local Marketplace Business—New Outlook in the Selected Developing Area

Nikola Vuksanović [1,*], Dunja Demirović Bajrami [2], Marko D. Petrović [2,3], Milan M. Radovanović [2], Slavica Malinović-Milićević [2], Adriana Radosavac [4], Valentina Obradović [5] and Maja Ergović Ravančić [5]

1  Faculty of Management, University "Union-Nikola Tesla", 21205 Sremski Karlovci, Serbia
2  Geographical Institute "Jovan Cvijić" SASA—Serbian Academy of Sciences and Arts, 11000 Belgrade, Serbia; d.demirovic@gi.sanu.ac.rs (D.D.B.); m.petrovic@gi.sanu.ac.rs (M.D.P.); m.radovanovic@gi.sanu.ac.rs (M.M.R.); s.malinovic-milicevic@gi.sanu.ac.rs (S.M.-M.)
3  Department of Regional Economics and Geography, Faculty of Economics, Peoples' Friendship University of Russia (RUDN University), Moscow 117198, Russia
4  Faculty of Applied Management, Economics and Finance, University Business Academy, 11000 Belgrade, Serbia; adriana.radosavac@mef.edu.rs
5  Faculty of Tourism and Rural Development in Požega, Josip Juraj Strossmayer University of Osijek, 34000 Požega, Croatia; vobradovic@ftrr.hr (V.O.); mergovic@ftrr.hr (M.E.R.)
*  Correspondence: vuksanovicnikola85@gmail.com

**Abstract:** Local producers constitute a crucial segment of the local economy, playing a pivotal role in driving rural development and the progress of tourism. Their avenues for showcasing products extend beyond markets, bazaars, or food events, often being integrated into a destination's tourism offerings. Moreover, they contribute to the culinary progress within tourism. Originating from wider rural areas or nearby villages, local producers significantly impact everyday migrations, services, and financial transactions in relationships spanning suburban–urban, village–town, and cross-border cooperation. This study aims to scrutinize the social facets of the organization and work of local producers, offering insights into contemporary market processes. It also serves to illustrate cross-border cooperation and the role of culinary tourism in local business. Through qualitative data processing, we will delve into the outcomes of cross-border projects, emphasizing ethical and sustainable values rooted in territory, landscape, local culture, authenticity, and the application of culinary elements in tourism. The results will shed light on the economic, social, and cultural ramifications on markets in border regions, influencing daily life and the economy. This study will define key aspects of rural development. These research findings can inform local governments, the economy, and communities in future strategic planning for developing this market segment. Tourism, especially in hospitality, will empower rural communities to enhance financial inflow and create local employment opportunities, such as roles for vineyard tour guides or local chefs. Simultaneously, it will bolster other sectors of the local economy, such as agriculture.

**Keywords:** rural development; culinary tourism; food; local producers; local economy; cross-border cooperation

## 1. Introduction

Within the realm of the research literature, individuals engaged in the production of local food and restaurants dedicated to showcasing regional cuisine are frequently recognized as lifestyle entrepreneurs. These entrepreneurs take pride in their produce and demonstrate a steadfast commitment to their local communities [1]. These characteristics are the central factors that contribute to an authentic gastronomic experience. Conversely, these local producers often operate on a smaller scale in terms of both workforce and financial turnover. Consequently, they may lack the resources to acquire essential market knowledge required to meet consumer demands. Another challenge faced by food producers can stem from difficulties in establishing sustainable retail relationships, potentially

arising from disagreements regarding pricing, volume, and delivery frequency [2,3]. For a lifestyle entrepreneur, achieving growth might not be the ultimate objective. This approach can result in a scenario where the actual availability of local products and meals becomes too restricted to adequately meet the demands of consumers [4,5].

The coordination of food events often involves the collaborative efforts of numerous participants, exemplified by food festivals, markets, and food fairs. Virtually every region hosts some form of food-related event, with a focus on either a specific local product, such as fish or apples, or a broader array of local offerings. These events may feature various competitions, such as those for the titles of chef, waiter, or sommelier of the year, or contests centered around specific products, like oyster opening championships or competitions for growing the largest pumpkin. The events vary in size, frequency, and target audience, ranging from being large-scale to intimate gatherings, recurring to one-time occurrences, and catering to both tourists and the local population or exclusively to locals. Some events are exclusively invitation-based. It is noteworthy that food events seldom operate as independent entities within the private sector; rather, local governments or associations commonly serve as the coordinating force, aligning the interests of diverse participants.

Through cross-border projects, local producers can achieve a lot of goals, among which the leading one is striving to promote sustainable economic and social development in border regions. Bearing in mind that these regions are usually passive and undeveloped, the projects of cross-border cooperation can stimulate the development of those activities in the local economy, which can result in better life conditions and economic prosperity on the one hand, and environmental protection on the other. In Serbia, instances of such projects primarily revolve around the advancement of rural and eco-tourism within specific regions afforded geographical protection [6].

Taking into consideration the mentioned facts, this paper examined the possibilities and perspectives of local producers as drivers of rural development and tourism progress, since the scarcity of studies and evident research gaps in this area existed. This research aims to focus on the social aspects of the organization and work of local producers, examining their actual conditions and management practices through the lens of a cross-border project in rural development and tourism advancement. This exploration aligns with the prevailing trends in the contemporary market process. The important aim is the identification of possibilities and perspectives in the management of market businesses conditioned by market trends with the use of qualitative methods such as interviews and observation of participants.

### 1.1. Gourmet Routes—Culinary Trails

Culinary tourism involves immersive travel experiences to gastronomically rich destinations, undertaken for recreational or entertainment purposes. This type of tourism encompasses visits to food producers, participation in food festivals, exploration of markets, engagement in culinary programs, and tasting of various food products, among other activities related to the culinary world. In recent years, culinary tourism has witnessed substantial growth, evolving into one of the most dynamic and creative segments within the broader tourism industry. More than one third of tourism consumption is dedicated to food. Moreover, culinary tourists face costs that are above average, and they are demanding and avoid uniformity. Therefore, gastronomy cannot become a mild and anonymous product—it has to have its own personality, because otherwise, it will become vulnerable, delocalized, and prone to forgery. Cross-border destinations often lack a comprehensive understanding of the importance of gastronomy in terms of diversifying tourism and catalyzing local, regional, and national economic development [7,8].

The success of food trails and culinary tours hinges on the collaboration of various stakeholders, including local producers, restaurants, hotels, wine producers, and breweries. These tours may be tailored to specific product types, such as beer, or encompass a broad spectrum of local products and dishes. Collectively, participants in this form of tourism can curate comprehensive tours or routes, featuring recommended stops where visitors

can gain insights into the stories behind the featured foods and meals. It provokes the emergence of questions of cooperation regarding such offers of experience. For example, culinary trails require certain common knowledge among the participants so that the food routes can be coherent experiences for tourists, and this can be challenging to achieve [9].

Collaborative endeavors between food and meal producers and local authorities have evolved into an integral facet of global destination marketing and development. The centralization of food and meal offerings in destination marketing is attributed to their broad appeal. Beyond attracting visitors with a keen interest in food [10,11], these offerings, even when not the primary motivation for travel, have the potential to convey broader experiential benefits associated with the destination, such as cultural impressions and insights. Moreover, culinary tourism embodies a spectrum of positive characteristics aligned with sustainable tourism [12], an increasingly valued aspect in destination development. These include the predominantly local economic impact of food tourism, the ecological advantage of reducing food miles in local food tourism, and the cultural benefits associated with the rediscovery and development of crops, cattle, food products, and meals. These cultural enhancements contribute positively to the local population's sense of cultural belonging and foster a better understanding of the visited place among tourists.

Culinary offerings and meals are intricately linked to places of diverse geographical scales, ranging from continents (e.g., Asian cuisine) and transnational regions (e.g., Mediterranean cuisine) to individual countries (e.g., Tastes of Sweden) and smaller national regions (e.g., Gourmet Bornholm). These gastronomic elements serve as key components for branding these locations to appeal to tourists. For instance, at a transnational level, collaborative efforts have been dedicated to refining and promoting the concept of New Nordic Cuisine. This culinary approach is rooted in utilizing seasonal products from the Nordic terroir, offering insights into the cultural heritage of Nordic ingredients and innovative applications of traditional Nordic foods [13]. However, establishing a destination, regardless of its size, as a recognized food destination in the international market is a formidable challenge due to intense competition from well-established culinary havens such as Italy, France, and Spain.

A crucial stride toward attaining recognition as a food destination involves fostering collaboration and coordination between the private and public sectors. This synergy is not only essential within each sector individually but also between the tourism and food sectors [14,15]. Although both public and private sectors may share the overarching objective of enhancing tourists' experiences with local cuisine, their approaches to achieving this goal can markedly differ. While local destination marketing organizations (DMOs) invest time in discerning diverse interests and negotiating compromises that cater to a broad spectrum of stakeholders, the private sector often prioritizes tangible outcomes for their business. While the public sector implements a strategic plan for collaboration, the private sector concurrently views organizations as having the same industry as potential competitors, potentially introducing challenges to broad-ranging cooperation.

*1.2. Culinary Events in Serbia*

Culinary events, both in the Republic of Serbia [16] and in the analyzed Vojvodina province (in the northern part of the country) [17], are held throughout the whole year. In accordance with the seasonality of food, traditional meals, desserts, or beverages, a lot of events are organized according to the content of that event.

Regarding the segments in culinary tourism, according to the results of the survey carried out for the needs of the "Global Report on Culinary Tourism" [18], in first place in attracting tourists are the events based on food offerings (79% of participants). They are followed by culinary trails (routes), classes, and workshops on the topic of gastronomy (62% of responses), then by food fairs of local products (59%), and visits to markets and producers (53%) [18]. Throughout the year, on the territory of Srem, numerous events are organized, where food and beverages, as well as general products of agricultural households from that area, have a significant role, which are shown in Table 1 [16].

**Table 1.** The selection of events organized on the territory of Srem.

| Month | Event—Festival | | |
|---|---|---|---|
| January | "Days of Wine" ("Dani vina") (Irig–Rivica) | | |
| February | "Srem sausage festival" ("Sremska kobasicijada") (Šid) | "Srem Wine Festival" ("Sremska vinarijada") (Šid–Berkasovo) | "The Return of Srem Despots" ("Povratak despota sremskih") (Pećinci–Kupinovo) |
| March | "And when saint's days pass, the cake remains" ("I kad prođu slave ostaje kolač") (Ruma) | "Srem Cake" ("Sremski kolač") (Ruma) | |
| April | Tourism Prism Stock Exchange (Berza turističke prizme) (Sremski Karlovci) | | |
| May | "FOOD FEST" Food and Beverage Festival (Sremska Mitrovica) | "Pećinci Fiacre Festival" ("Pećinačka fijakerijada") (Pećinci) | |
| June | Gugelhupf Festival (Festival kuglofa) (Sremski Karlovci) | "Srem Kulen Festival" ("Sremska kulenijada") (Erdevik–Šid) | |
| July | "Danube Festival" ("Dunavski festival") (Sremski Karlovci) | "Danube Festival" ("Dunavski festival") (Bačko Novo Selo) | |
| August | – | | |
| September | "Grapes Festival" ("Grožđebal") (Sremski Karlovci) | "Branko's Chain Dance" ("Brankovo kolo") (Sremski Karlovci–Stražilovo) | "Branko's Chain Dance" ("Brankovo kolo") (Novi Sad) |
| October | "Autumn is coming, my dear quince, and the corn is already ripe" ("Jesen ide, dunjo moja, kukuruzi već su zreli") (Irig–Jazak) | The Finals of "Zlatni kotlić Vojvodine" (Sremski Karlovci) | |
| November | – | | |
| December | "Karlovci Christmas Festivities" ("Karlovačke božične svečanosti") (Sremski Karlovci) | "Our Street" ("Naše sokače") (Inđija) | "Christmas Street" ("Božićna ulica") (Ruma) |

Wine tourism is getting more and more attention in our country. The practice has shown that tourists who go for wine routes are the consumers who support the wine region in many ways, directly through the consumption of not only wine, but also food specific to the visited region [19]. One of the eight wine regions of Serbia is the one in Srem with vineyards in Fruška Gora. The tradition of growing vines in Fruška Gora is about 1700 years old. The authentic wine of Srem is the already mentioned "bermet", which merchants exported to the USA 150 years ago. Today, Fruška Gora is known for its numerous kinds of wine with offerings from over 60 cellars of private wine producers, in Sremski Karlovci, Irig, Čerević, and Banoštor (Beočin), Erdevik (Šid), and Neštin (Bačka Palanka) [17,20].

*1.3. Overview of the Development of the Local Producers' Market*

The contemporary framework of food production and consumption has evolved within the principles of productivism [21], emphasizing profit, a global supply chain, and the exploitation of resources. This approach has given rise to a spectrum of consequences across social, economic, and ecological realms, which appear to have now reached a critical threshold. Numerous researchers concur that the current state of the "world of food" necessitates immediate interventions to rectify the trajectories of food systems. The existing food regimes have resulted in agriculture [22] becoming the least significant component within vast industrial mechanisms controlled by a small number of influential corporations. This trend is propelling us toward a state in which the depletion of soil, water, and biodiversity may soon become irreversible.

Thus, the alternative food network (AFN) is envisioned as a catalyst for transformative change toward more equitable, inclusive, and sustainable food systems. By reintegrating practices related to the production, distribution, and consumption of food both socially and spatially, these networks are perceived as having the potential to instigate profound shifts

in the food landscape. This transformation involves the establishment of local food systems that elevate the significance of small and family farmers employing environmentally sustainable methods. It also fosters more direct connections between these farmers and consumers, particularly those residing in urban areas [22].

Researchers have emphasized the considerable potential of the AFN in yielding valuable outcomes. Commonly reported benefits include positive local empowerment, economic sustainability, ecological sustainability, and social justice, with these elements often overlapping. The transition toward a localized food system is key to preventing the erosion of values by distant and transnational corporations. It revitalizes primary production sectors, particularly in peripheral regions [23]. Additionally, it serves as a foundation for more collaborative community development solutions, fostering trust and social capital that can extend to other collective initiatives. Moreover, it offers opportunities for synergy with other sectors, such as tourism [22].

From an economic standpoint, the sustainability of the AFN is rooted in consumers' ability to access fresh and healthy food at reasonable prices [24]. This, in turn, empowers producers by providing opportunities to increase profit margins [25], fostering diversification and entrepreneurship [26] and cultivating new skills [26]. Furthermore, the broader community can reap the benefits through various effects and the creation of new jobs in non-agricultural sectors [23]. On an ecological level, local initiatives for food production predominantly rely on organic and environmentally sustainable agricultural methods [27,28]. They derive ecological benefits associated with short supply chains, including the promotion of (agri)biodiversity, reduction in food miles, diminishment of carbon emissions, and advancement of sustainability objectives [22]. Regarding the characteristics of social justice of AFNs, researchers claim that the rebuilding of relations between participants brings them closer together and enhances mutual understanding [29] by encouraging respect, trust, and dedication [30], nurturing harmonious relations within communities and the participants' engagement in a more democratic food supplying [31].

Considering the aforementioned facts, local food strategies have taken center stage and are increasingly being implemented by local institutions. Across the globe, forward-thinking city governments are redefining their roles in national and international food systems. They are forging new connections between urban and rural areas by establishing innovative alliances between food producers and consumers. This presents an opportunity for cities to emerge as pivotal agents of change in food systems, contributing to the evolution of sustainable food practices [32]. This aligns with the recognition by Ref. [33] of a new paradigm grounded in ecosystems and territorial planning of food systems. The goal is not to supplant the global food supply chain but to enhance the local management of food systems, a perspective acknowledged by the FAO as inherently both global and local.

### 1.4. A Short Overview of the Serbian Case

Local producers and farmers have a chance to place their products mostly in markets, and then an opportunity appears in the form of food events, such as night bazaars [34] or hypermarkets, which are not the most popular. In Serbia's history, a marketplace served as the primary gathering spot for vendors and customers, playing a crucial role in the social life of the local population. However, in contemporary times, while markets continue to be significant meeting places offering agricultural and various other products, many producers have also opted for alternative trade channels [35]. Whichever channel of trade or distribution of products they choose, this exchange has a significant sociological aspect and social connection within their local community.

In Serbia, markets remain the predominant form of retail sales, serving as the primary mode of trade activity. As detailed in the literature, the wholesale trade of fruits and vegetables typically occurs at open-air "kvantaš" markets, where producers and vendors present their goods from trucks, trailers, or other vehicles. Nevertheless, in 2012, the Declaration of Cooperation for establishing a national wholesale market system was signed in Belgrade, the capital city of Serbia. Encompassing approximately 18 hectares, with

17,000 m$^2$ allocated for indoor space, the national wholesale market will feature 750 parking spaces for trucks, 500 parking spaces for customers, loading and unloading ramps, forklift services, waste recycling facilities, hospitality amenities, and financial institutions. The anticipated annual turnover is estimated to range between 800 thousand and a million tons of goods [36]. This example is proof that markets are the most accessible for local producers, but that there are also other trade channels such as bazaars, food events, and in the hospitality sector (restaurants, etc.).

*1.5. Review of the Local Cuisine of the Vojvodina Province (Northern Serbia)*

The richness of Vojvodina's home cuisine is most prominently showcased in its rural areas, where the diversity of its appeal arises from the coexistence of 24 nations on its territory. Among the most populous are Serbs, Hungarians, Yugoslavs, Croats, Montenegrins, Rusyns, Ukrainians, Poles, Germans, Albanians, Turks, Czechs, and Slovaks [37–39]. This amalgamation of cultures, religions, and nationalities forms the foundation of the local cuisine, constituting a key element in the offerings of rural tourism in Vojvodina. According to existing research [38–41], Vojvodina's home cuisine has predominantly evolved under the influences of Germany, Austria, Slovakia, and Hungary. Over the years, the people of Vojvodina have exchanged food customs and habits, culminating in the creation of a distinctive Vojvodina cuisine. The arrival of the Germans had a profound impact on food and its preparation, introducing their dishes, customs, pickling techniques, wine, fruit, and vine cultivation. Consequently, the Serbian population adopted many of their culinary practices. Moreover, there is a wealth of culinary knowledge stemming from interactions with Hungarian, Montenegrin, Ruthenian, and Slovakian cuisines. Numerous dishes are crafted in alignment with Croatian or other neighboring culinary traditions. Additionally, traces of Oriental cuisine, prevalent in Belgrade and other parts of Serbia, also influence the culinary landscape [42]. This amalgamation of diverse culinary influences renders Vojvodina cuisine distinct and inherently appealing to tourists. Despite these various influences, the dietary habits and customs of Vojvodina remain among the least changed, reflecting the enduring cultural life of its villages.

Given the outlined facts, the objective of this study is to examine the social aspects of the organization and operations of local producers. The aim is to gain insights into contemporary market dynamics and, specifically, to showcase the example of cross-border cooperation and its impact on the realm of culinary tourism within a specific geographical context.

## 2. Materials and Methods

Vojvodina, situated in the northern region of the Republic of Serbia, encompasses 24.3% of the country's total land area. As of the 2011 Census, the territory is home to 1,931,809 inhabitants [43]. Vojvodina is administratively organized into 7 districts, 45 municipalities, and comprises a total of 467 settlements, with 415 classified as rural [44].

Among the existing tourist routes in Europe, namely the West, Central, and East Mediterranean routes, the East Mediterranean route holds particular significance for Vojvodina. This route serves as a vital link connecting Northern, Central, and Western Europe with the Adriatic, Aegean, and Black Seas. It plays a crucial role in facilitating the movement of foreign tourists in Serbia, including Vojvodina, and in the ex-Yugoslav countries. Additionally, it is of importance for Greece, Romania, and Bulgaria. The primary tourist dispersion for Vojvodina encompasses the countries of Central, Western, and Northern Europe. However, following the separation of the Yugoslav republics, Vojvodina and Serbia faced new competition in the tourism market. Hungary, Croatia, and Slovenia emerged as significant competitors, particularly Hungary and Slovenia, which developed similar products in rural tourism as Vojvodina. This less favorable tourism–geographical position is a consequence of the western part of the former Yugoslavia, including Slovenia, Croatia, and Montenegro, separating into independent countries. Conversely, Vojvodina and Serbia

benefit from a more favorable position compared to countries in Eastern Europe, such as Bulgaria, Romania, and Greece.

Rural areas in Vojvodina confront several challenges, including a decline in the number of farmers (with the agricultural population comprising less than 11% of the total population), an increase in elderly households due to the migration of young people away from rural areas, and a reduction in agricultural land due to the expansion of industrial areas [45,46]. However, Vojvodina possesses substantial natural resources, abundant agricultural land, a traditional approach to agriculture, and considerable potential for rural tourism development. It also boasts recognizable traditional local food specialties and opportunities for the growth of complementary activities [47,48]. The term "Vojvodina cuisine" is particularly closely associated with rural areas. Although various nations, traditions, and cultures have influenced the local cuisine, the most significant impacts come from Hungary, the Ottoman Empire, and the Austrian–Hungarian Empire. The culinary traditions in Vojvodina, shaped by Hungarian and Ottoman influences, as well as those of the Austrian–Hungarian Empire, have been further influenced by Byzantine, Persian, and Arabic culinary elements throughout Vojvodina and the broader Balkan Peninsula [49].

The subject of this research is the cooperation on the cross-border project entitled "Authentic Gourmet Taste for Sustainable SocioEconomic development of the Cross-Border Region–Pannonia Gourmet", approved for funding from the Interreg-IPA CBC program, Croatia–Serbia 2014–2020. Even though the deadline for its completion was in 2021, due to the well-known COVID situation in 2020, the project was completed in 2022. The distinct objective of this project was to enhance, broaden, and unify the cross-border tourism offerings while also enhancing the management of cultural and natural heritage assets. The research areas were Croatian Syrmia, City of Vinkovci, and Serbian Syrmia, Municipality of Irig–Vrdnik (Figure 1).

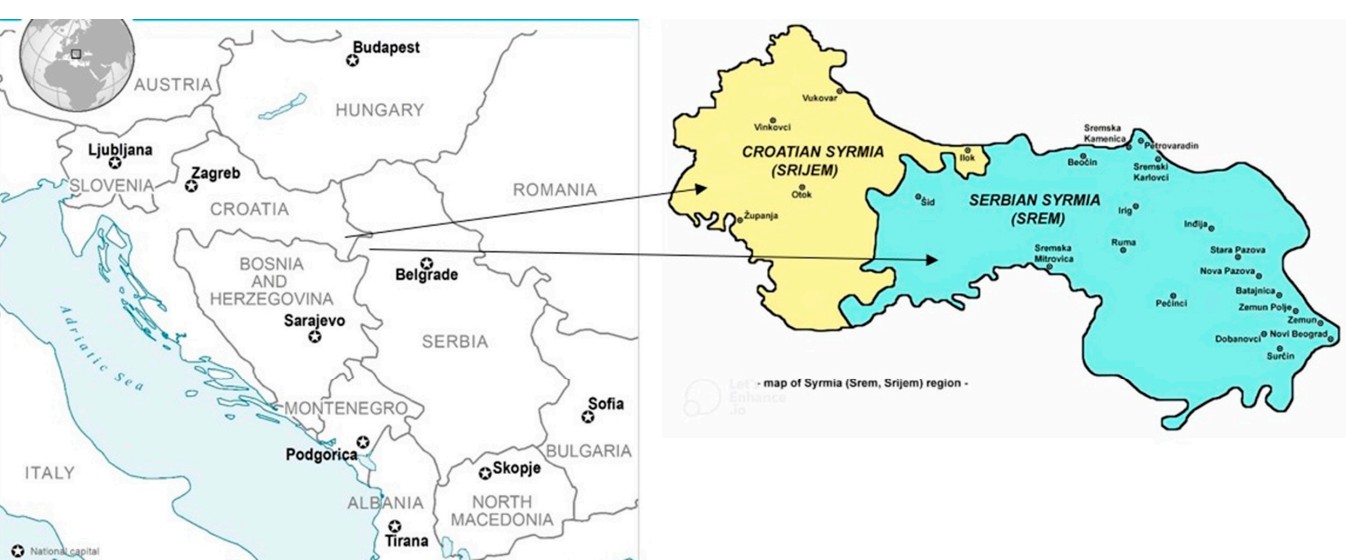

**Figure 1.** The position of study area. Source: [50].

The leading partner of this project was EDUCONS University [51], as well as the partners of the City of Vinkovci (Vinkovci, Croatia), Tourism Organization of the Municipality of Irig (Irig, Serbia), Vinkovci City Investment and Development Agency VIA Ltd. (Vinkovci, Croatia).

The overarching objective of this project was to establish the shared brand "Eat Pannonia (EP)" to promote newly developed tourism products. Local producers were the direct participants, and the cuisine of cross-border destinations played a pivotal role in enhancing the overall vacation experience. This project has brought together local producers, compiling authentic recipes that will be preserved and disseminated through two gourmet centers

located in Vrdnik and Vinkovci. Culinary tourism will serve as the primary motivation or a significant factor for visitors choosing to travel to this cross-border tourism destination.

What is the Eat Pannonia partnership?

*"The Eat Pannonia partnership is a network of tourism and hospitality services providers, small producers of food and beverages, culinary events, associations, institutions, and other bearers of tourism activities from the area of the Pannonian Plain"*.

*Sampling*

The initial phase of this research focused on the sociodemographic attributes of the sample. Interviews were conducted with 40 individuals, comprising 11 managers and 29 production owners. The interviewees exhibited diversity in terms of age, educational attainment, and residence. A majority were regular participants in bazaars and events, with half of them also engaging in markets. Over 50% of the production owners were couples (husbands and wives) of middle ages or older generations, predominantly possessing high school and university qualifications, and exhibiting a preference for urban areas.

More details about the sociodemographic characteristics of the interviewees are presented in Table 2. This study showed that this market matches the profile from other studies [49] because it retains its roles of opening jobs, wealth accruing, and catering to families of medium income, and it is a place that fosters the development of entrepreneurial skills.

**Table 2.** Sociodemographic characteristics of the interviewees.

|  | **Men** | **Women** |
|---|---|---|
| Total: | 27 | 13 |
| **Work position:** |  |  |
| Manager | 6 | 5 |
| Owner | 20 | 9 |
| **Age:** |  |  |
| ≤20 years | 0 | 0 |
| 21–30 years | 1 | 1 |
| 31–40 years | 4 | 1 |
| 41–50 years | 4 | 4 |
| 51–60 years | 9 | 4 |
| ≥61 years | 8 | 4 |
| **Activities:** |  |  |
| Bazaars | 30% |  |
| Food events | 30% |  |
| Local marketplaces | 40% |  |
| **Education:** |  |  |
| Elementary/Middle school | 1 | 1 |
| High school | 9 | 12 |
| Graduate degree | 11 | 6 |
| Postgraduate degree | 1 | 1 |
| **Residence:** |  |  |
| City | 1 | 1 |
| Surrounding areas | 13 | 16 |
| Other areas | 6 | 3 |

The selection of the local producers who would participate in the project was performed based on having the following criteria: active social networks, website, and mandatory activity in the food industry or in hospitality. According to the criteria, a team of experts in the fields of gastronomy, food product technology, enology, and tourismology

selected 40 local producers (Table 3). Other producers (18 of them) are in the process of joining, but they are equal members and participated in the research.

**Table 3.** The selection of the local producers.

| | Producer | Field (Food Industry or Hospitality) | Type of Product |
|---|---|---|---|
| 1. | But&Co (Laćarak, Serbia) | Food industry | Cured meat products |
| 2. | Akademska rakija (Novi Sad, Serbia) | Food industry | Strong alcoholic drinks |
| 3. | Razbeerbriga (Bukovac, Serbia) | Food industry | Craft beer |
| 4. | Mrkić (Veternik, Serbia) | Food industry | Honey |
| 5. | Marin špajz (Novi Sad, Serbia) | Food industry | Juices, jams, ajvar, and pickled salads |
| 6. | Truff Truff (Novi Sad, Serbia) | Food industry | Dairy products—cheese |
| 7. | Krivina (Novi Sad, Serbia) | Food industry | Strong alcoholic drinks |
| 8. | Žigon (Stari Banovci, Serbia) | Food industry | Strawberry products—jam and juice |
| 9. | Restaurant Lamut (Vinkovci, Croatia) | Hospitality | Food and beverage services |
| 10. | OPG Abelo (Vinkovci, Croatia) | Food industry | Strong alcoholic drinks based on honey |
| 11. | OPG Ana Delić (Župnja, Croatia) | Food industry | Cured meat products |
| 12. | Šokački stan (Vinkovci, Croatia) | Hospitality | Food and beverage services |
| 13. | OPG [1] Goran Ferbežar (Vinkovci, Croatia) | Food industry | Honey |
| 14. | Pastai (Sremska Kamenica, Servia) | Food industry and hospitality | Pasta production andfood and beverage services |
| 15. | Royal biscuits and decorations (Novi Sad, Serbia) | Food industry | Cookies |
| 16. | Restaurant SAT (Novi Sad, Serbia) | Hospitality | Food and beverage services |
| 17. | Katrica Biber (Sremski Karlovci, Serbia) | Food industry | Cake and sweets production (Gugelhupf and small gateaux) |
| 18. | Restaurant Alinea (Vinkovci, Croatia) | Hospitality | Food and beverage services |
| 19. | Restaurant ORION (Ivanec, Croatia) | Hospitality | Food and beverage services |
| 20. | Truff Truff BVS doo (Novi Sad, Serbia) | Food industry | Fungi products—truffles |
| 21. | Valentinijan Ltd. (Vinkovci, Croatia) | Food industry | Craft beer |
| 22. | Vigus ltd (Vinkovci, Croatia) | Food industry | Cake production |

Source: https://eatpannonia.rs/#/clients-page (accessed on 12 January 2023). [1] OPG—Obiteljsko poljoprivredno gospodarstvo "Family agricultural holding".

Qualitative data collection was obtained in several waves during 2020. Interviews were carried out with the production owners or managers who showed interest in participating in this research.

This study involved a population of 40 participants (sample size). For qualitative research, sample sizes of 10 participants can be deemed sufficient when sampling from a homogeneous population [52]. Similar interpretations regarding sample size for quantitative research are found in the literature [53–56]. In this context, the sample size in this research (N = 40) was considered adequate for conducting reasonable assessments. Therefore, qualitative methods were chosen for collecting and analyzing the presented data. This selection aligns with the contemporary trend of studying vendors' and customers' behavior in the market field [56–59]. As Creswell [53] notes, this type of interview is commonly employed in various social studies to gather data from interviewees, facilitating the evaluation of highly idiosyncratic phenomena alongside structured methods focused on specific topics.

The interview procedure was developed with a theoretical foundation, encompassing two primary areas: (1) the sociodemographic profile of the interviewees and (2) general inquiries about their work and organization on the market. At the outset, the interviewees were briefed on the research subject, and they were assured of the research's anonymity. All pertinent statements were recorded in the Serbian language to preserve them as original statements, facilitating thorough analysis with minimal loss of information.

Moreover, all pertinent data obtained from the interviews, crucial for the analysis, underwent transcription and translation into the English language. This process included the systematic organization and categorization of the discourse. The collected data were subsequently processed using the open-source software KH Coder. Building upon theoretical assumptions and participant observations, an elucidation of the key questions within the discourse for each interviewee was conducted, along with the identification of patterns. The research technique employed was grounded in observational research, involving the direct observation of the phenomena under investigation in their natural environment.

The objective of this study is to investigate the social aspects of local producers' organization and operations, examining their status and management within the context of cross-border cooperation. Given the active nature of similar research projects in this part of Europe [6] and the Iberian regions [60], this study aims to make a valuable contribution to the existing literature. Moreover, this research holds significance as it constitutes a crucial component in the advancement of culinary tourism. It sheds light on the social organization of local producers, contemporary market processes, decision-making dynamics, and market outcomes in Serbia.

## 3. Results

With the 40 interviews conducted, we were able to identify and analyze differences and similarities between the interviewees' points of view which facilitated the creation of a general line of thought. Overall, the topic of the "Eat Pannonia" brand was clearly implemented in every production owner's or manager's mind at different scales and through different behaviors.

Using a word frequency list (Table 4) and a word cloud (Figure 2) as the initial tools for this research, we could easily visualize the most frequently used words by the interviewees which enabled us to make relevant conclusions.

Considering the word frequency list (Table 4), the general tendency of the production owners or managers is represented by single words that gained importance acting individually. All these words, despite having an important role independently, relate with each other in a crucial way.

**Table 4.** Word frequency list.

|  | Word | POS/Conj. | Frequency |
|---|---|---|---|
| 1 | Food | Noun | 39 |
| 2 | Quality | Noun | 32 |
| 3 | Product | Noun | 30 |
| 4 | Natural | Noun | 28 |
| 5 | Homemade/grown | Noun | 25 |
| 6 | Authentic | Noun | 23 |
| 7 | Agriculture | Noun | 21 |
| 8 | Traditional | Noun | 20 |
| 9 | Rural/Local | Noun | 20 |
| 10 | Healthy | Noun | 18 |
| 11 | Tastes | Noun | 17 |

Source: Self-elaborated based on KH Coder 3.

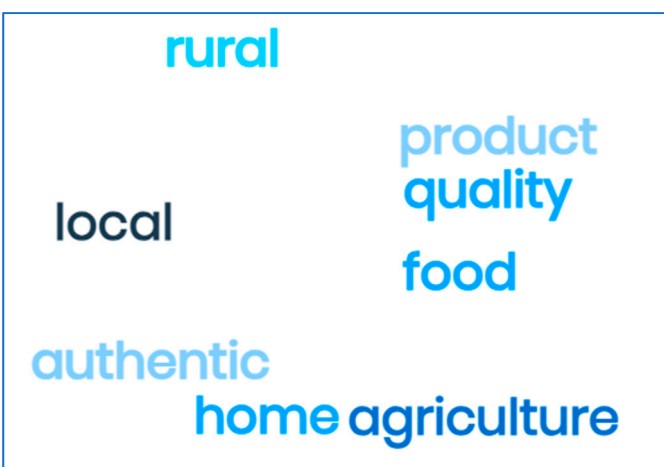

**Figure 2.** Word cloud. Source: Self-elaborated based on Tag Crowd.

The primary finding, as depicted in Table 4, was derived from the prevalent terms used by interviewees when defining the significance of the "Eat Pannonia" brand. A majority of respondents associated the brand with a product, particularly emphasizing the quality of food as a "natural product" (86% of responses). This quality is linked to the (in)direct interaction with the producer, facilitating the provision of fresh products to customers. The second most frequently employed concepts include "homemade/-grown" and "authentic" (69% of responses), followed by "agriculture" (54% of responses), "traditional" (52% of responses), and "rural" (51% of responses). Additional concepts within this context encompass "return to tradition", "harmonization", "healthy food", and "tastes and smells", each receiving less than 50% of responses. One interviewee aptly described the "Eat Pannonia" brand as symbolizing the "traditional environment," while another emphasized its role in the "harmonization of the Pannonian oasis".

Due to the contextual and functional similarity of both Table 4 and Figure 3, the analysis was conducted together. In general, all respondents had the same attitude when asked about the importance of the "Eat Pannonia" brand from a general perspective in their business. Despite being the focus of this research, the words "food", "product", and "quality" were undoubtedly a relevant topic that was analyzed. This turned out to be an important or even fundamental aspect. Words like "local" and "agricultural" were obviously common because of the general theme. "Agricultural" was introduced in the contexts of "home" and "authentic". Accordingly, the word "local" appeared as one of the most frequently used, because the local product is at the center of this research and is crucial to culinary application in tourism.

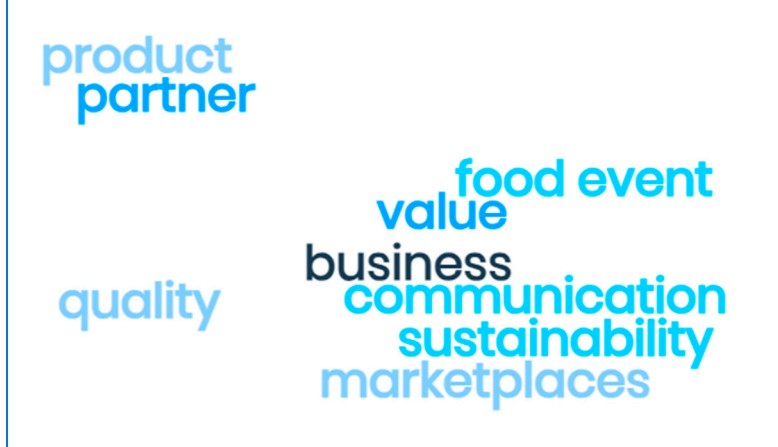

**Figure 3.** Word cloud. Source: Self-elaborated based on Tag Crowd.

Using the word frequency list (Table 5) and the word cloud (Figure 4) as initial tools for this research, we could again easily visualize the most frequently words used by the interviewees which again enabled us to make relevant conclusions.

**Table 5.** Word frequency list.

|  | Word | POS/Conj. | Frequency |
|---|---|---|---|
| 1 | Partner | Noun | 39 |
| 2 | Quality | Noun | 32 |
| 3 | Communication | Noun | 30 |
| 4 | Value | Noun | 28 |
| 5 | Product | Noun | 25 |
| 6 | Business | Noun | 23 |
| 7 | Sustainability | Noun | 21 |
| 8 | Traditional | Noun | 20 |
| 9 | Marketplaces | Noun | 20 |
| 10 | Food events | Noun | 18 |
| 11 | Local | Noun | 17 |

Source: Self-elaborated based on KH Coder 3.

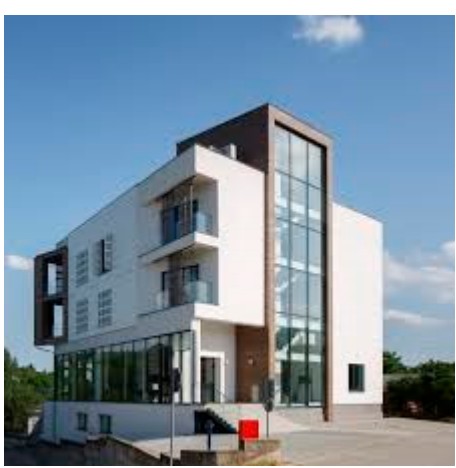

**Figure 4.** Gourmet Center, Vrdnik [61].

In addition to better understanding the product characteristics, we also gained insights into the nature of interactions and contributions within the partnerships associated with the "Eat Pannonia" brand. Numerous respondents described "partner interaction" (41% of the respondents) as being related to communication with producers and the expansion of their "product value", and one of the respondents specifically said "that the contact with customers is mainly traditional, and that entering the tourism sector provides new possibilities and perspectives for the development, which is a mission of our business". This shows the importance of communication in business because it seems that besides the warm and close interpersonal contact that they have in marketplaces, bazaars, and food events, new business opportunities may appear. These terms generally reveal the importance of challenges and the decisions for market expansion. These findings emphasize the importance of the manifold function of the market in terms of places, the activities of local producers, and the mitigations of social problems present in everyday life.

Considering other important words presented in the Figure 4, we noticed that the words "food event", "business", "communication", "sustainability", "value", and "marketplaces" were clearly interconnected and represented their importance to production

owners or managers. This was unanimous between all interviewees. Production owners or managers have a notion of their own potential for development and implementation in culinary tourism.

The insights derived from the most frequently mentioned terms revolve around overlapping elements of positive local connection and economic sustainability. This type of product migration primarily serves to prevent the concentration of values in the hands of large companies, thereby mitigating the marginalization of the production sector, especially in peripheral areas. This study aligns with findings in other research [22] and lays the groundwork for potential collective solutions in community development. It also contributes to fostering trust and social capital, which can extend to other collective and community initiatives. Furthermore, it presents opportunities for creating synergy with other sectors, such as culinary tourism.

### 3.1. Social Organization/Components of the Observed Market

Another facet of the research findings pertained to the overall aspects of the social organization of the market. The primary social components observed were linked to the frequency of participants' visits to the selected locations and their observation of daily events.

The interactions between everyday work activities in production and contact with the place of product delivery are noticeable. Moreover, the current work activities and decisions are connected with delivery. Many of them have their own delivery vehicles and the previously agreed upon terms for delivery, both to the reseller and to their places of sales. Supplier–producer–customer interactions are carried out in marketplaces on weekdays, and at weekends, at food events, while once a month, they go to the night bazaar [34].

Regarding the inquiry about weekly visits and interactions with customers, numerous respondents characterized them as a "positive approach" and regular communication in the form of trust. As one of the respondents said, their daily sales vary every day. Now, the new possibility to deliver goods to the Gourmet Center develops production and also changes the logistics of the product migration, as remarked by one of the participants.

Many of the participants stressed that this form of doing business broadened their perspective and possibilities for their presence, which was very important both for them and for those who go to marketplaces, bazaars, etc. Considering all of the above, production owners or managers have a notion of their own potential for development and implementation in culinary tourism.

### 3.2. Contemporary Processes of the Observed Market—Gourmet Center

Within this project, an educational center–the Gourmet Center–was built (Figure 5). Its main aim is to be an active promoter of culinary tourism offerings of the cross-border area, which will enable the networking of the key actors of the sector, increase their capacities, as well as help in the preservation of authentic food and recipes. Also, the center is open to tourists and the scientific community for the needs of organizing various private and business events.

The center is equipped with a professional kitchen with adequate educational (Figure 6), exhibition, and tasting spaces, as well as with accommodations for the needs of the realization of project activities.

Furthermore, through this project, a common Map of Eat Pannonia (EP) partners was created, which enables the digital presentation of their products and services in a unified and efficient way. At the same time, a modern ICT platform and mobile application were created as an innovative solution for the promotion of culinary tourism offerings in the cross-border area, and it will have the following content:

- Eat Pannonia Map with a possibility of creating a route;
- Quality assessment of the presented culinary tourism offerings in the area;
- Up-to-date thematic news from the region.

Based on the information of EP partners and products, and the standards of branding EP, one map will be designed and created. All EP partners who have been certified for branding their products will be marked and described in detail. The map will provide information on the location of the producers and their products. The map will be further used as a visual presentation of ICT solutions developed during the project implementation.

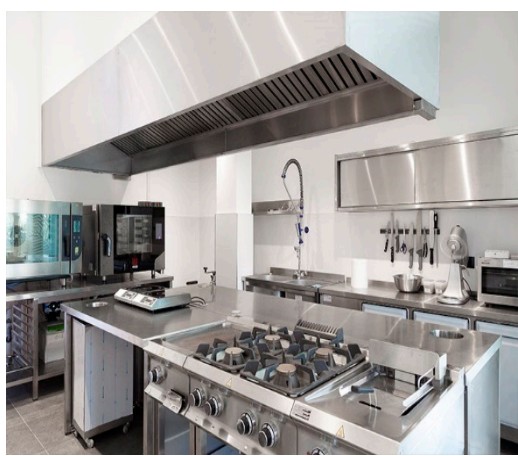

**Figure 5.** Professional kitchen [61].

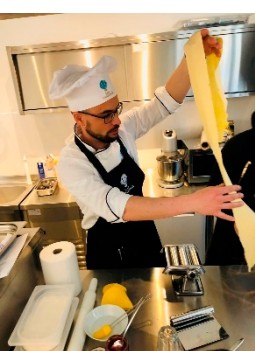

**Figure 6.** Making homemade fresh pasta (Author photo).

The processes and decisions were defined for the admission to the EP community, and the following interested parties can become partners:

- Food and beverage producers;
- Hospitality facilities;
- Tourism organizations;
- Tourist agencies;
- Team building agencies;
- Culinary events;
- Citizens' associations/Clusters/Cooperatives;
- Local action groups;
- Other companies and organizations from the private, civil, and public sectors who express their wish to join and meet the general and specific criteria defined by the rulebook on using the sign.

The EP menu was selected by a team of experts in the fields of gastronomy (where one of the authors was the team manager), food technology, tourism, and enology. The result of the mentioned research was the Book of Methodological Standards with the Guidebook for the Eat Pannonia brand, which comprises old Pannonian recipes (Table 6), ways of preparation, and ingredients from that area, which make those dishes different and unique.

**Table 6.** Traditional recipes from the areas of Srem and Srijem.

| Type of Dish | Name of Dish | Ingredients |
| --- | --- | --- |
| Starters | Eat Pannonia board | Cured meat products |
| | Seoski čekić—Rustic hammer | Bacon and prunes |
| | Homemade bread with raspberries and goat cheese | Bread, raspberry jam, and goat cheese |
| | Goose liver pate (Pâté de foie gras) | Goose liver |
| | Leba, mast i paprika—Bread, fat spread, and paprika powder | Bread, fat spread, and paprika powder |
| Soups and broths | Banat soup | Guinea fowl and root vegetables |
| | Kikinda broth | Pumpkin and root vegetables |
| Main dishes | Goose stew | Geese—the whole trunk |
| | Pork shank in dark beer | Pork shank and dark beer |
| | Veal in milk | Veal |
| | Lovački gulaš—Hunter's stew | Game meat |
| | Bačka—Style perch | Perch |
| | Drunken carp | Carp |
| | Winter magic | Sausages made of Mangalica meat |
| Salads | Monastery salad | Pickled cabbage, walnuts, cream, and horseradish |
| | Baked pepper with fresh cheese | Long pepper and fresh cheese |
| | Lettuce with hazelnuts and grapes | Lettuce, hazelnut oil, and grapes |
| Desserts | Gomboce sa šljivama—Plum dumplings | Potato dough and fresh plums |
| | Jabuka u šlafroku—Apple in a robe | Apples and sweet dough wrapping "orli" |
| | Salenjaci—Lard cakes | Flour, lard, and jam |
| | Rezanci sa makom—Noodles with poppy seeds | Homemade pasta and poppy seeds |

One of the authors himself also participated (Figures 6 and 7) in the active promotion of culinary tourism offerings in the cross-border area, which enabled the networking of the key actors in the sector, the growth of their capacities, as well as the preservation of authentic food and recipes and is crucial for culinary application in tourism.

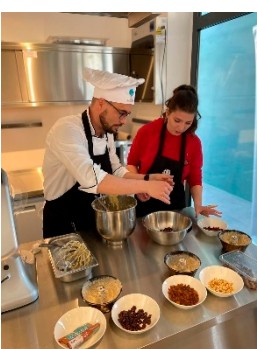

**Figure 7.** Preparation for Gugelhupf (Author photo).

Taking into consideration the mentioned facts, it can be concluded that the direct participants are local producers whose products will be presented on the map of culinary tours, sold as edible souvenirs, and used in the production of certain traditional meals and, they are the main factor for culinary application in tourism.

## 4. Discussion

In the context of safeguarding natural and cultural heritage for developmental purposes, this research points to the local management of natural resources and presents a popular choice of activities which have the aims of preserving biodiversity, cultural identity, and economic growth, as well as the developing through service-providing activities, primarily tourism. This is in line with Petrović et al. [47], who argued that if people are aware of the local management of natural resources, it will present a popular choice of activities.

When customers are observed as the primary actors, the farmers in the local system of food can consequently be regarded as reactive. Actually, like a typical/idealized neoliberal subject, the participation of local producers in local food events can only be a response to the market forces caused by the customers' demand for local food and positive attributes connected with it. This aligns with Halkier [4] and Hal et al. [5], who asserted that growth may not always be the ultimate objective. This situation can result in a limited range of local products and meals, which might not adequately meet consumers' needs.

Through projects involving cross-border cooperation, many goals can be reached, one of the leading ones being the striving for the promotion of sustainable economic and social development in cross-border regions. Knowing that these regions are usually passive and undeveloped, projects involving cross-border cooperation can foster the development of these activities in the local economy, which can provide better life conditions and economic prosperity on the one hand, and the protection of the environment, on the other. This concept aligns with the perspective of Đukić et al. [6], who posited that in Serbia, such projects predominantly focus on the development of rural and eco-tourism in specific regions with geographical protection.

Taking into consideration the abovementioned facts, the statements give an insight into the contemporary market process and point to the competitive advantages and the potential for the improvement in culinary tourism of the territory of Vojvodina. The statements point to the current position of the sector of culinary tourism and they provided an insight into the desired development in the future. Gathering information "in the field", as well as gathering data from secondary sources, enabled us to gain insight into the current situation in the sectors of agriculture, market, and socioeconomic status. Mapping cross-border resources offers a new outcome of this aspect of doing business for the interested parties.

The role of culinary tourism in local business also depends on the cooperation of numerous participants, such as local producers, restaurants, hotels, wine producers, and breweries, and they can be directed toward certain types of products, such as beer, or toward a wide range of products (local products or dishes). Collectively, participants in this form of tourism [62] can provide an encompassing tour or route with suggested stops, allowing visitors to gain insights into the narrative behind each food/meal. Additionally, culinary tourism plays a significant role with various positive attributes related to sustainable tourism [12], which is becoming more appreciated in destination development. These include the economic impact of culinary tourism predominantly benefiting local businesses and events. Furthermore, local culinary tourism and food products contribute positively to fostering a sense of cultural belonging among the local population.

Following the analysis of data and discussions on the results, several key findings will be highlighted as the following:

- Production owners or managers recognize the overall importance of associating a brand with a product, particularly when it is crucial for culinary applications in tourism.
- For most of the respondents, the significance of the "Eat Pannonia" brand primarily relates to the "quality of food". The second most frequently used concepts include "homemade/-grown", "authentic", "agriculture", "traditional", and "rural".
- From the overall perspective of their businesses, the "Eat Pannonia" brand emerged as a significant or even fundamental aspect for most respondents. Agriculture was introduced in the contexts of home and authenticity, with "local" being one of the

most frequently used terms. This emphasis on local products is crucial for culinary applications in tourism, which is the focal point of this research.

- The terms "food event", "business", "communication", "sustainability", "value", and "marketplaces" are clearly interconnected and represent the importance of production owners or managers.
- The term "partner interaction", for most of the respondents, is related to communication with consumers and the expansion of the "product value".
- For most of the respondents, entering the tourism sector provides new possibilities and perspectives for the development of culinary application in tourism.

## 5. Conclusions

The research findings emphasize the significance of culinary tourism for local businesses and highlight the potential for implementing programs in the development of cross-border cooperation, involving agriculture, producers, tourism, and rural sectors. This form of cross-border collaboration implies joint investments in infrastructure, spatial organization, environmental protection, and the collective presentation of tourism products to third-party markets. The nexus between food and tourism can serve as a foundation for local economic development. Experiences in consuming homemade food and beverages contribute to shaping a destination's brand, promoting local culture, making it attractive to tourists, and facilitating the integration of culinary applications in tourism.

In essence, the integration of development perspectives from consumers, producers, and destinations enhances the comprehensive understanding of culinary tourism. This approach considers various interests and leverages the resources of producers across different sectors and public–private collaborations, with potential contributions from consumers. Such a collaborative perspective among stakeholders proves highly beneficial for both the practical development of culinary tourism and the conceptual broadening of this field.

These findings could influence future research on different market aspects, contingent on regional disparities, and offer alternative avenues for local development in less advanced communities. Local authorities, the economy, and communities could leverage the insights from this research for future strategic planning in developing this market segment. These findings highlight the economic, social, and cultural impact on the markets of the cross-border region, influencing the daily lives of people and the economy. Additionally, it aids in defining the aspects of rural development.

Tourism, especially hospitality tourism, will enable rural communities to improve their fund inflow and provide them with the opportunity for employment at the local level by offering jobs for vineyard tour guides and for local chefs, and it will also foster other sectors of the local economy, such as agriculture.

Tourism holds the promise of boosting local agricultural growth by creating connections that allow nearby farmers to meet the food demands of tourism spots. Strengthening these connections is crucial in fostering a cooperative relationship rather than a conflicting one between agriculture and tourism. Drawing them closer together brings advantages such as reducing reliance on imports, enhancing the quality of food supplied to the tourism industry, giving tourists better access to local cuisine, and promoting sustainability in tourism while also tackling poverty. Bridging agriculture and tourism is a way to prevent economic losses and spread the benefits of tourism throughout local communities. Essentially, by understanding what consumers want, local entrepreneurs can effectively use tourism as a catalyst for investing in new economic ventures by strengthening ties with the tourism industry.

In the realm of tourism, this idea emphasizes the role of chefs and hospitality managers as vital cultural mediators who bridge the gap between local and tourist dining experiences. Cuisine serves as a lens through which we can discern the power dynamics among tourists, tourism enterprises, and the local community, showcasing whose cultural preferences take precedence in the presentation of food. Importantly, it underscores how tourism businesses significantly influence local food-producing communities by shaping

the culinary landscape—an essential consideration, especially in recognizing the growing role of businesses as drivers of development.

**Author Contributions:** Conceptualization, N.V. and M.D.P.; methodology, D.D.B.; software, S.M.-M.; validation, M.E.R. and V.O.; formal analysis, M.M.R.; investigation, N.V.; resources, N.V.; data curation, D.D.B.; writing—original draft preparation, N.V. and M.D.P.; writing—review and editing, A.R.; visualization, M.E.R. and V.O.; supervision, M.M.R.; project administration, A.R. All authors have read and agreed to the published version of the manuscript.

**Funding:** This research received no external funding.

**Institutional Review Board Statement:** Not applicable.

**Data Availability Statement:** The data presented in this study are available on request from the corresponding author.

**Conflicts of Interest:** The authors declare no conflicts of interest.

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
