# Peer review of "The Role of Culinary Tourism in Local Marketplace Business—New Outlook in the Selected Developing Area"

_agriculture, doi:10.3390/agriculture14010130_

Round 1

Reviewer 1 Report

Comments and Suggestions for Authors

I would recommend a number of upgrades.

1.The authors must improve the readability and understandability of paper. The topics and issues are well explained but the paper is "dense" and full of details. It would be helpful for the reader to be helped by Tables and other forms of summarizing the information to highlight the main insights of the paper. 

2.The factor "culinary/gastronomy" is underplayed under a long list of subjects. It is not clear the role to be assigned the culinary topic. Please be more explicit about this factor.

3.Some of the information provided may not be relevant. Example: Table 2: Traditional recipes.

4.It would useful to provide a Table with the main topics identified by the software KH Coder.

5.Rewrite the paper to offer a more sound, coherent and focused approach. Most of the information is relevant, but must be framed and contextualised within the boundaries of the paper´s title and main theme. 

Reviewer 2 Report

Comments and Suggestions for Authors

The Discussion paragraph, as such, needs to be improved. Findings are not faced with before research. No references at all are found in this part.

The first paragraph of the part of results should be included in the Methodology part.

The findings are not shown clearly in the paper. 

Also, the manuscript needs to use, in a higher way, tables, and maps, to clarify the study area, or the findings, i.e.

Round 2

Reviewer 1 Report

Comments and Suggestions for Authors

The authors solved most of the problems. Still think that the papers contains too much information, but within acceptable limits. Congratulations.

Comments on the Quality of English Language

No major issues.

Reviewer 2 Report

Comments and Suggestions for Authors

The authors have revised the manuscript according to the suggestions. Therefore, the proposal deserves to be published.